# Not simply a matter of parents—Infants' sleep-wake patterns are associated with their regularity of eating

**Christophe Mühlematter[1], Dennis S. Nielsen[2], Josue L. Castro-Mejía[2], Steven A. Brown[3†], Björn Rasch[1], Kenneth P. Wright, Jr.[4], Jean-Claude Walser[5], Sarah F. Schoch[6☉], Salome Kurth** [1,7☉] *

1 Department of Psychology, University of Fribourg, Fribourg, Switzerland, 2 Department of Food Science, Faculty of Science, University of Copenhagen, Copenhagen, Denmark, 3 Institute of Pharmacology and Toxicology, University of Zurich, Zurich, Switzerland, 4 Department of Integrative Physiology, University of Colorado Boulder, Boulder, Colorado, United States of America, 5 Genetic Diversity Center, ETH Zurich, Zurich, Switzerland, 6 Donders Institute for Brain, Radboud University Medical Center, Nijmegen, Netherlands, 7 Department of Pulmonology, University Hospital Zurich, Zurich, Switzerland

☉ These authors contributed equally to this work.

† Deceased.

* salome.kurth@unifr.ch

**Data Availability Statement:** Ethical restrictions (cantonal ethics committee of Zurich, Switzerland) apply to this paper, which prevents the public sharing of individual data that contain potentially

## Abstract

In adults there are indications that regular eating patterns are related to better sleep quality. During early development, sleep and eating habits experience major maturational transitions. Further, the bacterial landscape of the gut microbiota undergoes a rapid increase in complexity. Yet little is known about the association between sleep, eating patterns and the gut microbiota. We first hypothesized that higher eating regularity is associated with more mature sleep patterns, and second, that this association is mediated by the maturational status of the gut microbiota. To test this hypothesis, we performed a longitudinal study in 162 infants to assess actigraphy, diaries of sleep and eating times, and stool microbiota composition at ages 3, 6 and 12 months. To comprehensively capture infants' habitual sleep-wake patterns, 5 sleep composites that characterize infants' sleep habits across multiple days in their home environment were computed. To assess timing of eating habits, we developed an Eating Regularity Index (ERI). Gut microbial composition was assessed by 16S rRNA gene amplicon sequencing, and its maturation was assessed based on alpha diversity, bacterial maturation index, and enterotype. First, our results demonstrate that increased eating regularity (higher ERI) in infants is associated with less time spent awake during the night (sleep fragmentation) and more regular sleep patterns. Second, the associations of ERI with sleep evolve with age. Third, the link between infant sleep and ERI remains significant when controlling for parents' subjectively rated importance of structuring their infant's eating and sleeping times. Finally, the gut microbial maturational markers did not account for the link between infant's sleep patterns and ERI. Thus, infants who eat more regularly have more mature sleep patterns, which is independent of the maturational status of their gut microbiota. Interventions targeting infant eating rhythm thus constitute a simple, ready-to-use anchor to improve sleep quality.

sensitive information. Participants giving consent to the study were not informed that their and their children's data, might be shared publicly. Data are available from (contact: salome.kurth@unifr.ch, info.kek@kek.zh.ch, and Tobias.Rosenberger@kek.zh.ch) for researchers who meet the criteria for access to confidential data.

**Funding:** We acknowledge funding from the Swiss National Science Foundation (PCEFP1-181279 to SK; P0ZHP1-178697 and P2ZHP1_195248 to SS) and the University of Zurich (Medical Faculty; Forschungskredit FK-18-047, Clinical Research Priority Program "Sleep and Health", to SK) Foundation for Research in Science and the Humanities (STWF-17-008, to SK), and the Olga Mayenfisch Stiftung (to SK). KPW was supported by the Office of Naval Research (ONR) Multiple University Research Initiative (MURI) Grant N00014-15-1-2809 and NIH R56 HL165343. The funders had no role in study design, data collection and analysis, decision to publish, or preparation of the manuscript.

**Competing interests:** KPW reports research support/donated materials from commercial entities DuPont Nutrition & Biosciences; Grain Processing Corporation; and Friesland Campina Innovation Centre. KPW has served on the scientific advisory board and received consulting fees from commercial entity Circadian Therapeutics, Ltd; has served on the scientific advisory board without consulting fees from commercial entity Circadian Biotherapies, Inc., and as a consultant without consulting fees from US government agency the United States Army Medical Research and Materiel Command – Walter Reed Army Institute of Research. This does not alter our adherence to PLOS ONE policies on sharing data and materials.

# Introduction

Infants' sleep-wake patterns vary greatly, much more than at any other stage of life [1] and these variations can be influenced by different factors such as diet [2] or microbes residing in the digestive tract, known as the gut microbiota [3]. Yet little is known about the interplay between sleep, eating and the gut microbiota. Recently, the possible pathways have been reviewed of how gut microbiota characteristics are intertwined with diverse aspects of sleep [4]. Depletion of the gut microbiota through intake of antibiotics leads to changes in sleep quality [5] and sleep architecture [6–8]. The gut-sleep interaction is likely bidirectional, such that experimental sleep fragmentation alters gut microbiota profiles in rodents [9–12]. Consequently, factors that modify the gut microbiota might indirectly also affect sleep.

The primary factor determining gut microbial composition is diet [13]. Yet, evidence suggests that also the timing of food intake affects gut microbiota and—when regular—supporting the diurnal dynamic in gut bacterial abundance [14–16]. Relatedly, defined time windows of eating and fasting are interconnected with sleep. Experimental restriction of caloric intake to a consistent time window during the day in *Drosophila* increases total sleep duration and reduces the diurnal distribution of sleep [17]. In adults with metabolic syndrome, time-restricted eating positively affects subjective sleep quality as well [18, 19].

The positive effect exerted by time-restricted eating might originate from rhythmic gut microbiota fluctuations, and/or from the support of physiological rhythmicity. The developmental transitions across the first year of life in sleep patterns [1, 20], circadian rhythms [21], and the complex ecosystem of gut microbiota [22, 23], may likely interact with food intake regularity. During this time, the diversity of the gut bacterial community increases and a shift in the prevalence from being dominated by *Bifidobacterium* to *Bacteroides* occurs in many infants [3, 22]. Our previous findings link daytime sleep and bacterial diversity, nighttime sleep fragmentation and variability with bacterial maturity and enterotype [3]. Infant eating rhythms have not yet been examined as part of this concept.

Thus, the first goal of this analysis was to determine the relationship between eating regularity and sleep patterns in healthy infants. During infancy the meals are frequent and scattered throughout day and night. Based on the developmental patterns that infants become increasingly rhythmic overall, we hypothesized that increased eating regularity relates to more mature sleep patterns, as quantified with five "sleep composites" that comprehensively capture infants' sleep-wake patterns [20]. Specifically, we expected that more regular eating relates to less time spent awake at night, shorter sleep duration during the day, a longer sleep period at night, earlier bedtimes, and more regular sleep-wake patterns. Second, we hypothesized that associations between sleep and eating regularity are mediated by the maturational status of the gut microbiota, as quantified with three markers of infant gut bacterial maturation: alpha diversity determined as Observed species, Shannon, and Chao1 indexes, bacterial maturation index, and enterotypes [3]. Third, parenting principles might influence infants' sleeping and eating schedules, confounding the association between eating regularity and sleep. Accordingly, we hypothesized that the association of eating and sleep extends beyond the importance parents give to regular infant schedules, assessed through the score of Structure from the Baby Care Questionnaire [24].

# Methods

## Study population

In Switzerland, parents of 162 healthy infants were recruited in 2016–2019 for participation in longitudinal assessments at infant ages 3, 6 and 12 months. Infants were generally healthy,

born by vaginal delivery, and were fed at least 50% through breast milk at enrollment. Exclusion criteria were taking antibiotics or medication affecting their sleep, or having chronic diseases or disorders of the central nervous system. Participants are described in more detail in Schoch et al. [20], which included the same dataset for analysis. Subject data were coded with an identifier code number. This project was approved by the cantonal ethics committee of Zurich, Switzerland (BASEC 2016–00730) in accordance with the Helsinki Declaration. Before enrollment, parents received explanations on the study procedure and gave written consent.

## Sleep

At the three assessment timepoints (3, 6, 12 months), sleep-wake patterns were derived from movement patterns measured with acceleration sensors (i.e., actigraphy, GENEactiv by Activinsights LtD, Kimbolton, UK, 43x40x13mm, MEMS sensor, 16g, 30 Hz Frequency) for 11 consecutive days. The device was worn on the left ankle of infants attached with a paper strap or in a sock designed to hold it. The actimeter was worn continuously and only removed for baths. Each removal of the device was documented by caregivers in a detailed 24-h diary [25, 26]. In addition, manual documentation also included sleep/wake phases, external movement during sleep (e.g., stroller), phases of crying, and the clock timing of when meals started and ended.

Combining the data from actimetry and 24-h diary, sleep and wake periods were identified following our laboratory standards and sleep composites were computed [20, 25]. The use of sleep composites was decided to greatly reduce the number of sleep variables, thereby minimizing the risk of type I errors [27]. Moreover, our objective was to comprehensively explore the associations between eating regularity and sleep as opposed to selecting single sleep variables that would underestimate the individual and transient dynamics of infants' sleep patterns. From 32 sleep variables we compiled 5 sleep composites: **Sleep Activity** (lower values refer to higher sleep efficiency, less movement during sleep, and shorter nocturnal wake episodes), **Sleep Day** (lower values refer to decreased number of naps and longer wake time during the day), **Sleep Night** (lower values refer to shorter time between bedtime and get up time), **Sleep Timing** (lower values refer to earlier bedtime) and **Sleep Variability** (lower values refer to reduced variability in bedtime, get up time or sleep duration between measurement days).

## Maturational markers of gut microbiota

Stool sample DNA extraction and gut microbiota characterization by 16S rRNA gene amplicon sequencing was performed as described in detail in Schoch et al., (2022) (see supplementary information). We used alpha diversity, bacterial maturation index and enterotype as markers of gut microbiota maturation. In alignment with our previous work, we used the alpha diversity measures Observed species, Shannon, and Chao1 [3]. For bacterial maturation index, a random forest analysis was computed on the bacterial composition [28] to predict the maturity of each sample, which was then compared to the actual age at the assessment timepoint. This provided a relative bacterial maturation index, with positive values indicating a comparably more mature bacterial profile within the sample and vice versa for negative values [3]. Enterotypes classify individuals' gut microbial composition and assign them to groups with similar bacterial profiles. Here a Calinksi–Harabasz 2-cluster based approach using weighted UniFrac metrics as input were used and highest Calinksi–Harabasz value and high prediction strength were computed and the samples assigned to either a *Bifidobacterium*-rich (A) or a *Bacteroides*-rich enterotype (B) [3, 29].

## Eating Regularity Index

To characterize the regularity of infant day-to-day food intake timing, and based on the concept of the Sleep Regularity Index [30], we developed an Eating Regularity Index (ERI) with scores from 0 to 1. Parents reported the clock time of infants' meals across the assessment days in a 15-min epochs resolution. For each subject, clock times were transformed into a vector of 96 data points for each assessment day (*i.e.*, 15-min epochs contained in a 24-h-day). This binary vector equaled 1 at clock-times when a meal started or 0 when this was not the case.

We applied a simple moving average (SMA) to the vectors twice, so that the epochs neighboring the one when the meal occurred would be attributed a value depending on its distance to the meal, with closer epochs having higher values. We applied the SMA interval of 4 as it revealed highest ERI between-subjects variability (measured as standard deviation) at 6 and 12 months, and was close to the highest variability at 3 months (S2 Fig). The similarity between the vectors adjusted with the double SMA was measured using cosine similarity computed for every possible pair of days within each subject at each assessment—reflecting the likelihood of eating at the approximate same time within all recording days. The mean of these paired comparisons defines the core measure of eating regularity ERI with values from 0 (highly irregular) to 1 (highly regular).

Because the Eating Regularity Index (ERI) relies on consistency across multiple assessment days, subjects with fewer than 5 days of continuous sleep diary were excluded from the analysis (n = 7, 1 at 3 months, 3 at 6 months, and 3 at 12 months).

## Parenting principles

To control for the parental differences in terms of eating and sleeping schedules imposed on the infants, we included an in-lab translated German version of the Baby Care Questionnaire as a control variable [24], with 30 parent-rated items on a 4-points Likert scale (strongly disagree, disagree, agree, strongly agree). This questionnaire provides a score "Structure", representing parents' subjective importance to introducing a schedule in their infant's day (for example concerning regular food timing or bedtimes). Thus, the score "Structure" was included as a covariate in each analysis computing the association between the ERI and sleep composites to assess parental influence as potential cofounder linking sleep and eating regularity.

## Statistical analysis

For statistical analysis we used R version 4.0.5 with packages proxy [31], zoo [32], dplyr [33], suncalc [34], lubridate [35], multilevel [36], mice [37] and broom.mixed [38]. Plots were created by using R package ggplot2 [39]. In each model, we included sex (0 for male, 1 for female), exact age, breastfeeding (0 for not or rarely breastfed, 1 for occasionally, regularly, or daily breastfeeding), and the mean number of meals per day during the assessment period as control variables. Since infants tend to eat less during nighttime, the number of meals taken between 7 am and 7 pm divided by the daily number of meals was also included as a control variable.

To test the association between the ERI and sleep, we ran five multilevel models (one for each sleep composite as the outcome with the ERI as an independent variable). The Akaike Information Criterion and the Bayesian Information Criterion were used to determine the need for a random interval and random slope for each multilevel model. For Sleep Variability, Sleep Timing, Sleep Day, and Sleep Activity, each participant had a random intercept. For the model with Sleep Night, a random slope was added to the random intercept for each participant. To assess age-related dynamics within the eating regularity and sleep construct, we computed generalized linear models for the three age groups separately (3, 6, and 12 months).

Breastfeeding was not included in the model for 3 months old as all infants were primarily breastfed at this age.

Finally, to evaluate whether gut microbiota mediate the relationship between the ERI and sleep we computed mediation models by using the Baron & Kenny 4-step Causal Analysis approach [40]. Gut microbiota markers were included as mediator, i.e., alpha diversity (Observed, Chao1, Shannon), bacterial maturation index, and enterotypes, and the ERI was implemented as independent variable, while the five sleep composites represented dependent variables. Relative bacterial phyla level abundance was also explored as a possible mediator between ERI and sleep composites. Similar to the multilevel sleep models, number of meals, ratio of meals between 7 am and 7 pm, exact age, sex, breastfeeding, and the Structure score of the Baby Care Questionnaire were used as control variables. The alpha level was set to $P < 0.05$.

## Results

### Eating regularity increases with age

First, we examined whether eating becomes more regular across the first year of life. Indeed, the ERI experienced an age-related increase (F(2, 443) = 14.98, p < 0.001; One-way ANOVA, Fig 1); accompanied by a decrease in the number of meals with age (S1 Fig). From 3 to 6 months, the ERI did not increase significantly (p = 0.11, 95% C.I. = -0.004, 0.052; Tukey post-hoc test), but it significantly increased between 3 and 12 months (p < 0.001, 95% C.I. = 0037 0.095). The ERI increased also from 6 to 12 months (p < .001, 95% C.I. = 0.013, 0.071; Tukey post-hoc tests).

### Infants' eating regularity is associated with sleep variability, sleep quality and bedtimes

We then tested the first hypothesis whether the ERI is associated with infants' sleep patterns (Fig 2). We observed that a higher ERI was associated with lower Sleep Variability (t(393.639) = -6.694, b = 2.007, p<0.001), earlier Sleep Timing (t(392.554) = -2.658, b = 3.468, p = 0.008)

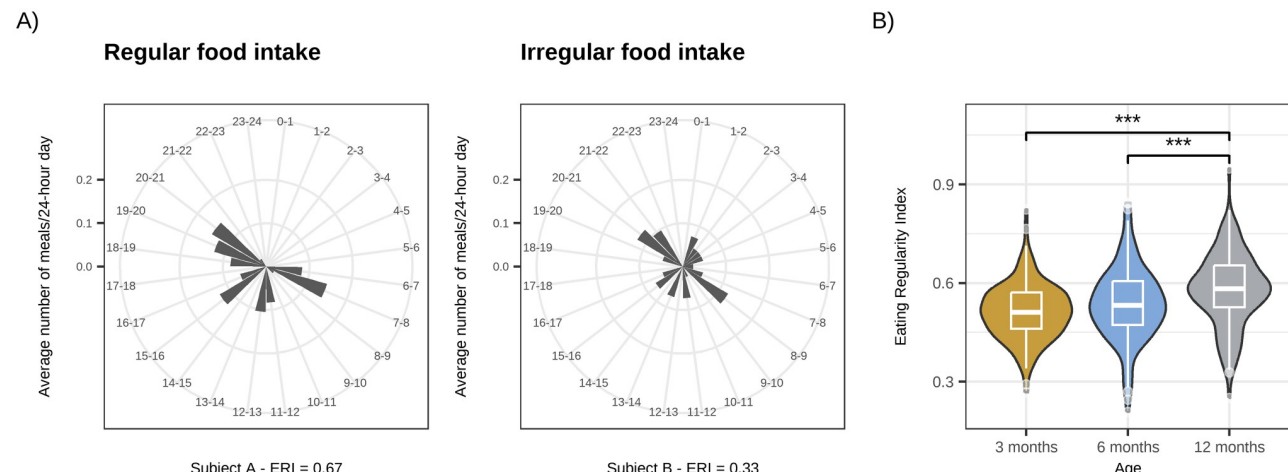

**Fig 1.** A) Representative examples of food intake timing regularity across the day for two infants at age 12 months with a high (left) and low (right) Eating Regularity Index (ERI). The y-axis refers to the relative number of meals taken at corresponding clock times. B) ERI at ages 3 (n = 149), 6 (n = 153) and 12 (n = 140) months. The ERI increases with age (p < 0.001; One-way ANOVA), yet, not significantly between 3 and 6, but between 3 and 12 months and between 6 and 12 months (*** p < 0.001).

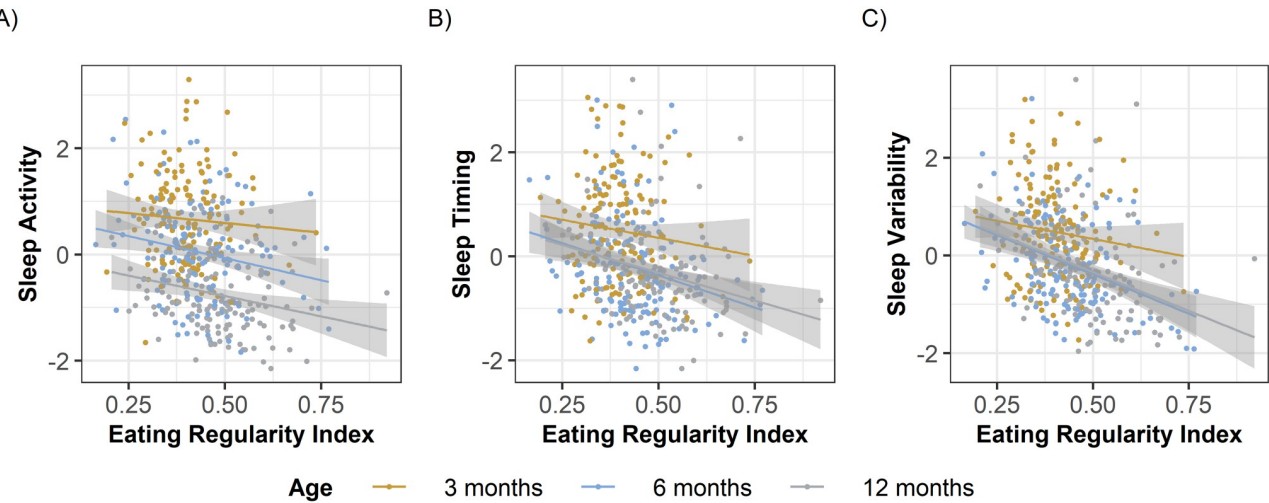

**Fig 2. Association of eating regularity (ERI) and infant sleep composites.** The association between the ERI with Sleep Activity, Sleep Timing, and Sleep Variability. A) The ERI was significantly associated with Sleep Activity when all ages are combined (multilevel model across all ages, p = 0.001), at 12 months (p = .005) and 6 months (p = 0.016), but not at 3 months (p = 0.43). B) The ERI was significantly associated with Sleep Timing when all ages are combined (multilevel model across all ages, p = 0.008), at 12 months (p = 0.029) but not at 6 months (p = 0.055) and at 3 (p = 0.143). C) The ERI was significantly associated with Sleep Variability with all ages pooled (multilevel model across all ages, p <0.001) and significant effects at all ages: 3 months (p = 0.002), 6 months (p<0.001), and 12 months (p < .0001).

and lower Sleep Activity (t(391.042) = -3.22, b = 1.739, p = 0.001). In other words, infants with more regular food intake had more regular sleep, went to sleep earlier, and woke up less frequently during the night. Yet, contrary to our hypothesis, the ERI was not associated with the sleep composites relating to the duration of sleep during night or day: no significant association was found between ERI and Sleep Day (t(381.787) = 0.279, b = 3.059, p = 0.781) and between ERI and Sleep Night (t(383.344) = -0.521, b = -2.83, p = 0.603).

To investigate if the associations among ERI and infant sleep changed with age, we assessed each age group separately. Although generalized linear models revealed that Sleep Activity was significantly associated with ERI at 12 months (t(123.356) = -2.874, b = 6.874, p = 0.005), and at 6 months (t(137.229) = -2.442, b = -0.879, p = 0.016), such associations were lacking at 3 months (t(119.021) = -0.791, b = 1.818, p = 0.43, Fig 2A). In other words, the linkage between ERI and Sleep Activity emerged between age 3 and 6 months. The link between the ERI and Sleep Timing was only significant at 12 months (t(122.574) = -2.216, b = 4.239, p = 0.029), and lacking at 3 or 6 months (3 months: t(120.092) = -1.475, b = 4.837, p = 0.143, p = .12; 6 months: t(140.37) = -1.937, b = 6.275, p = 0.055, Fig 2B). Thus, the association between ERI and Sleep Timing emerged between age 6 and 12 months.

The link between ERI and Sleep Variability was stable across the first year of life, with significant effects at all ages: 3 months (t(118.489) = -3.247, b = 1.969, p = 0.002), 6 months (t (134.794) = -5.218, b = 1.188, p<0.001) and 12 months (t(124.63) = -4.624, b = 8.677, p<0.001, Fig 2C).

The associations between ERI, Sleep Night and Sleep Day were not significant for any age group (all p>0.05, S1 Table).

## Parental structure association with infant's sleep

The Baby Care Questionnaire score of Structure, a measure for the importance given by parents to introducing schedules in their infant's day, was included in the multilevel models as

a control variable. Parents' score of Structure was significantly linked with Sleep Timing (t (400.876) = -5.501, b = 3.468, p<0.001), Sleep Day (t(391.533) = -3.15, b = 3.059, p = 0.002) and Sleep Night (t(396.675) = 3.098, b = -2.83, p = 0.002). There was no significant link with Sleep Variability (t(389.556) = -1.917, b = 2.007, p = 0.056) nor with Sleep Activity (t(400.831) = 0.027, b = 1.739, p = 0.979). There was an association between Parents' score of Structure and their infant's ERI (t(413.973) = 3.062, b = -0.142, p = 0.002).

The general linear models showed that parental Structure was consistently associated with Sleep Timing at all ages (3 months: t(120.112) = -2.818, b = 4.837, p = 0.006; 6 months: t (139.44) = -4.355, b = 6.275, p<0.001; 12 months: t(124.864) = -2.56, b = 4.239, p = 0.012). Its association with Sleep Night was only observed at 3 months (t(119.867) = 2.755, b = -4.703, p = 0.007) and 6 months (t(138.605) = 3.563, b = -3.611, p<0.001), but not at 12 months (t (124.473) = 1.01, b = -3.042, p = 0.315). No association of parental Structure with Sleep Day was found at 3 months (t(115.577) = -0.697, b = 6.027, p = 0.487), nor at 6 months (t(130.873) = -1.884, b = 2.429, p = 0.062). However, it was significant at 12 months (t(118.408) = -2.149, b = 3.386, p = 0.034).

## The link between eating regularity and infant sleep does not appear to be mediated by gut microbiota diversity and profile

Finally, we tested whether maturational markers of gut microbiota relate to the association of ERI and infant sleep, i.e., alpha diversity, enterotype, and bacterial maturation index. As expected, alpha diversity increased across age in Observed, Chao1 and Shannon indices (S3 Fig). We performed a mediation analysis to assess the role of the microbial markers in the association between ERI and sleep. Building on the first step of this Causal Mediation Analysis (i.e., Fig 2), we computed the second step including the association between ERI and the three gut microbial markers, which revealed a positive association of ERI with alpha diversity, indicating that infants who ate more regularly showed increased microbial diversity. This association was significant for the Observed species index (t(360.706) = 2.348, b = 67.212, p = 0.019), Chao1 index (t(360.679) = 2.56, b = 92.77, p = 0.011), and for Shannon index (t(369.327) = 2.201, b = 1.082, p = 0.028). The mediation analysis rejected the second hypothesis, revealing that alpha diversity is neither a significant mediator in the model with Sleep Activity (Observed: t(354.161) = 0.429, b = 1.926, p = .66; Chao1: t(365.282) = 0.353, b = 1.926, p = 0.72; Shannon: t(374.097) = 0.079, b = 1.945, p = 0.93), nor Sleep Variability (Observed: t (383.156) = -0.504, b = 2.548, p = .61; Chao1: t(378.24) = -0.67, b = 2.569, p = .50; Shannon: t (390.406) = -0.656, b = 2.57, p = 0.51), nor Sleep Timing (Observed: t(372.636) = 0.178, b = 3.592, p = 0.85; Chao1: t(366.624) = -0.193, b = 3.614, p = 0.84; Shannon: t(384.228) = 0.335, b = 3.577, p = 0.73). Therefore, the full mediation models were not computed.

Similarly, we tested the association between the ERI and bacterial maturation index, which was not significant (t(350.959) = 1.354, b = 1.043, p = 0.17). Finally, the mediation analysis focused on the enterotypes, which were not associated with the ERI (t(391.384) = -0.397, b = -0.128, p = 0.69), suggesting no mediation effect of enterotype in the ERI-sleep association. Similar to alpha diversity, full mediation models for bacterial maturation index and enterotype could not be computed.

Finally, we explored the association of eating regularity with specific microbial signatures at the phyla level, which revealed that the ERI is associated with the relative abundance of *Firmicutes* (t(405.937) = 3.379, b = -0.002, p<0.001, Fig 3), but not with any of the other phyla (S2 Table). To investigate how the associations among ERI and *Firmicutes* abundance changed with age, we assessed each age group separately. Generalized linear models showed that the ERI is associated with *Firmicutes* relative abundance at 12 months (t(128.032) = 2.718,

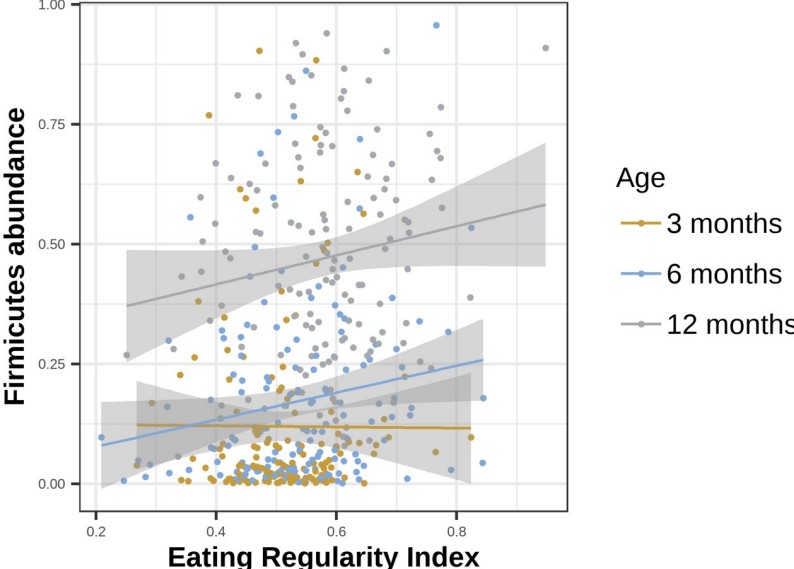

**Fig 3. Association of eating regularity (ERI) and *Firmicutes* relative abundance.** The ERI was significantly associated with Firmicutes relative abundance when all ages are combined (multilevel model across all ages, p<0.001), at 12 months (p = 0.007), but not at 6 months (p = 0.07), nor at 3 months (p = 0.78).

b = 1.487, p = 0.007), yet not at 6 months (t(137.307) = 1.783, b = -0.383, p = 0.07) nor 3 months (t(121.036) = 0.27, b = 0.038, p = 0.78). Next, examined whether *Firmicutes* relative abundance mediate the ERI-sleep link, which revealed no mediation for any of the sleep composites (p>0.1).

Thus, neither the selected maturational markers nor phyla of infant gut microbiota are mediators of the association between ERI and sleep, yet linkages are found between ERI and microbial phyla in this cross-sectional study.

## Discussion

Eating and gut microbiota patterns have been associated with sleep patterns in animals and adult humans. This relationship likely emerges in early life when sleep regulation is being established. In a longitudinal study, we examined whether regularity of eating is associated with infants' sleep patterns across the first year of life and whether this association is mediated by the maturation of the gut microbiota. This investigation revealed four primary findings. First, infants eating more regularly exhibit lower variability in day-to-day sleep patterns, earlier bedtimes, and less fragmented nighttime sleep. Thus, we show that eating regularity is associated with more mature sleep patterns during infancy. Second, the links between eating regularity, sleep fragmentation, and bedtimes are emerging processes that appear between 3 and 6 months for sleep fragmentation and between 6 and 12 months for bedtimes. In contrast, eating regularity consistently relates to day-to-day sleep opportunity variability across the first year of life. Third, these observed associations remain significant even when including parental structure (i.e, the importance of giving a structure to their child's day) as a covariate. Fourth, our results do not support the hypothesis that changes in simple gut microbiota maturational markers mediate the connection between eating regularity and sleep patterns in infants. The findings do not rule out maturational changes in functional microbiome outcomes being associated with sleep maturation, which should be examined in future studies. Thus, our findings

confirm the concept that eating regularity is primarily associated with more mature sleep patterns during infancy.

We used multilevel models to test the association between infants' regularity of eating, quantified by our newly developed ERI, and the five sleep composites that comprehensively characterize infants' sleep habits across multiple days in their home environment [20]. In line with our hypothesis, the ERI positively correlated with decreased Sleep Variability and Sleep Activity, and higher Sleep Timing. More regular eating was not linked to sleep duration during night or daytime. Our results align with previous research, which reported that adults who eat irregularly also have comparably worse sleep quality [41]. Yet, research on adults in their seventies [42] did not demonstrate a link between eating regularity and sleep quality, thus this association may devolve with advancing age. Experimental studies implementing time-restricted eating demonstrate that setting a consistent time frame for eating (in this particular case overnight fasting) increased subjective sleep quality in adults [19, 43]. Moreover, time-restricted eating also affected diurnal/nocturnal sleep distribution and increased total sleep in *Drosophila* [17]. Restricting food access to a given time frame during the day might thus indirectly promote more regular eating times. Future studies might benefit from including a measure of eating regularity (like the ERI) within the imposed eating time frame. To our knowledge, our study is the first to demonstrate that eating regularity in infants is associated with more than just sleep quality (Sleep Activity). Furthermore, eating regularity was also associated with earlier bedtimes (Sleep Timing) and lower variance of sleep opportunity (Sleep Variability). Thus, similar to imposing a rhythmic light regimen which positively affects health outcomes in preterm-born infants [44], increasing eating regularity might prove a novel way to foster healthy sleep and support development.

To explore how the relationship between eating patterns and sleep evolves with age, linear models were computed for age 3, 6 and 12-months separately. This analysis demonstrated that even though Sleep Variability was significantly linked to the ERI at all ages, the association with Sleep Activity reached significance only at 6 months, while for Sleep Timing it reached a trend at 6 months and became significant at 12 months. It is important to note that the regularity of meals seems to be related to the regularity of sleeping times early in development, visible in that Sleep Variability being already correlated with the ERI at 3 months of age. It has been proposed that regular eating times support circadian rhythmicity [19]. Thus, regular eating times might act as an exogenous signal to enhance the development of circadian rhythms in infants, and especially the peripheral clockwork [45]. Yet, the link between eating and sleep regularity might be the result of the maturation of the hypothalamus, giving more rhythmic signals to both sleep and hunger [46]. The age-dependent effect of ERI on sleep timing and consolidation might also be explained by the evolution of circadian rhythm as a developmental transition. Indeed, different physiological measures such as cortisol, temperature or melatonin begin to show circadian rhythmicity simultaneously [21, 47, 48]. Thus, the link between eating regularity and these variables might rely on physiologically evolving rhythms. Future research is needed to clarify the mechanisms underlying the linkage of sleep and eating rhythm.

Our hypothesis primarily examined the impact of eating regularity on sleep patterns, however it is important to acknowledge that this association is likely to be bidirectional (confirmed in a multiple regression model with significance of the three sleep composites as predictors—Sleep Activity, Sleep Timing and Sleep Variability—and the ERI as dependent variable, data not shown). Prior studies have indicated that sleep restriction can lead to increased dietary intake in both preschool children and adults [49, 50], suggesting that sleep duration also influences appetite regulation. Hence, it is crucial for future research to further investigate the directionality of the association between eating behavior and sleeping patterns during development in order to gain a better understanding of this relationship.

Parents have a crucial influence on their infant's eating and sleeping times. We thus captured subjective parental ratings of establishing daily schedules, for example concerning bedtimes or mealtimes [24]. In line with our hypothesis, results revealed that the association between eating regularity and sleep patterns is not only driven by parent-imposed regularity. Interestingly, our model also showed that increased scores of parenting "Structure" related to earlier bedtimes (Sleep Timing), less sleep during the day (Sleep Day), and more during the night (Sleep Night). In other words, our results demonstrate that infants of parents who impose a daily structure slept more during the night, less during the day, and went to bed earlier. Our findings show that the extent of parental structure does not relate to infant's variability of sleep-wake patterns, nor to night awakenings, hinting that this might depend more on other factors. Indeed, in older children, a link was reported between increased parental concern about their child's sleep and children's worsened sleep quality [51, 52]. Thus, structuring young children's schedules can affect the diurnal distribution of sleep and bedtimes, but their sleep quality is linked with other factors such as eating regularity or parental cognitions on children's sleep.

One goal of this study was to test if the association between sleep patterns and eating regularity is mediated by selected microbial markers. Contrary to our hypothesis: the maturational status of the gut microbiota, as quantified with alpha diversity (determined as Observed species, Shannon, and Chao1 indexes), bacterial maturation index, and enterotypes does not appear to mediate the link between eating regularity and sleep patterns. However, the ERI was significantly associated with Observed and Chao1 indexes of alpha diversity. With an exploratory analysis of microbial phyla, we went beyond diversity measures. The phyla that differed most between subjects with high and low ERI (i.e. *Firmicutes*) did not drive the sleep-ERI association, yet the relative abundance of *Firmicutes* correlated with the ERI.

The association between eating regularity and sleep might be mediated by other aspects of the gut microbiota, e.g., its dynamics in composition across the 24-h day. Indeed, the timing of food consumption influences the diurnal changes in abundance and function of the gut microbiota in adults [15]. Interestingly, abundance of the phylum *Firmicutes* in mice is dynamic throughout the day and driven by fasting and feeding patterns, such that higher abundance is observed during feeding periods [16]. Thus, regular eating times might upscale *Firmicutes* relative abundance also in infants, and thereby support the microbial rhythm. Yet the cross-sectional measures of the gut microbiota used in our study cannot demonstrate intra-individual rhythmicity. Nonetheless, it is likely that a regular eating pattern in infants also supports the formation of rhythmicity in the gut microbial profile. Whether gut rhythms explain the association between eating regularity and sleep remains to be examined with repeated intra-individually collected samples.

An alternative explanation of the link between eating regularity and sleep would be that this association does not directly relate to the dynamics of the gut microbiota, but instead to metabolites influencing the microbiome, such as melatonin. It has been shown that the gut cells synthesize melatonin depending on the timing of feeding [53], which in turn can affect different gut bacteria gene expression [54]. The biosynthesis of melatonin by the gut could also have an effect through the peripheral circulation, but it is not yet known if the biosynthesis of melatonin by the gut cells has an endocrine effect [55]. Relatedly, eating sends information to the brain through vagal afferent nerves, which regulate hormonal secretion [56]. This was demonstrated in a rodent model with suprachiasmatic lesion, where restricted feeding successfully served as a zeitgeber to restore melatonin rhythmicity [57]. Thus, eating regularity may act through vagal afferent signaling, independent from gut microbiota. As eating times can be used as a zeitgeber, regular eating patterns might help support the development of physiological rhythms, which might then indirectly affect sleep. So even though eating times affect gut

microbial rhythmicity, it might also have an effect on sleep-wake regulation through the endocrine system.

## Conclusion

This study revealed that during early infancy, eating at regular times is associated with sleep quality, bedtimes, and variability of the sleep opportunity. Moreover, we show that the association of eating regularity with sleep patterns does neither purely result from parental behavior nor gut microbiota maturation. Even though future research must specify the directionality of this link, timing of eating is a potential variable to improve sleep quality early in life. Even though the explicit mechanism underlying this linkage remains to be fully elucidated, these novel results may have clinically relevance, such that introducing regular eating schedules could be used to support health development in infants at-risk. Notably, eating regularity is easily modified and future research is needed to determine if individually tailored rhythms could help to ameliorate problematic sleep-wake patterns by reducing sleep fragmentation.

## Supporting information

**S1 Protocol. DNA extraction from stool and processing of gut microbiota.**
(DOCX)

**S1 Fig. Number of meals evolution with age.**
(TIF)

**S2 Fig. Determination of simple moving average interval.**
(TIF)

**S3 Fig. Evolution of bacterial diversity from 3 to 12 months.**
(TIF)

**S1 Table. Sleep composites association with Eating Regularity Index.** Results of multilevel models and general linear models with the sleep composites as dependent variable and the Eating Regularity Index as an independent variable.
(DOCX)

**S2 Table. Phyla association with Eating Regularity Index.** Results of multilevel models and general linear models with the different Phyla as dependent variable and the Eating Regularity Index as independent variable.
(DOCX)

## Acknowledgments

We thank the parents and infants for participating in our study. We would like to thank the students and interns of the Baby Sleep Laboratory for their help with data collection. Additionally, we thank Andjela Markovic, Matthieu Beaugrand, and Valeria Jaramillo for reviewing analysis code, general discussion and feedback.

## Author Contributions

**Conceptualization:** Christophe Mühlematter, Dennis S. Nielsen, Josue L. Castro-Mejía, Steven A. Brown, Björn Rasch, Kenneth P. Wright, Jr., Sarah F. Schoch, Salome Kurth.

**Data curation:** Sarah F. Schoch.

**Formal analysis:** Christophe Mühlematter, Jean-Claude Walser, Sarah F. Schoch.

**Funding acquisition:** Sarah F. Schoch, Salome Kurth.

**Investigation:** Christophe Mühlematter, Dennis S. Nielsen, Josue L. Castro-Mejía, Steven A. Brown, Björn Rasch, Kenneth P. Wright, Jr., Sarah F. Schoch, Salome Kurth.

**Methodology:** Christophe Mühlematter, Dennis S. Nielsen, Josue L. Castro-Mejía, Jean-Claude Walser, Sarah F. Schoch, Salome Kurth.

**Project administration:** Christophe Mühlematter, Salome Kurth.

**Resources:** Salome Kurth.

**Software:** Christophe Mühlematter, Sarah F. Schoch.

**Supervision:** Sarah F. Schoch, Salome Kurth.

**Visualization:** Christophe Mühlematter.

**Writing – original draft:** Christophe Mühlematter, Salome Kurth.

**Writing – review & editing:** Christophe Mühlematter, Dennis S. Nielsen, Josue L. Castro-Mejía, Steven A. Brown, Björn Rasch, Kenneth P. Wright, Jr., Sarah F. Schoch, Salome Kurth.

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
