## [Decision Letter · Decision Letter 0]

19 Jun 2023

PONE-D-23-13344Not simply a matter of parents - Infants’ sleep-wake patterns are associated with their regularity of eatingPLOS ONE

Dear Dr. Kurth,

Thank you for submitting your manuscript to PLOS ONE. After careful consideration, we feel that it has merit but does not fully meet PLOS ONE’s publication criteria as it currently stands. Therefore, we invite you to submit a revised version of the manuscript that addresses the points raised during the review process.

We look forward to receiving your revised manuscript.

Kind regards,

Takahiro J. Nakamura, Ph.D.

Academic Editor

PLOS ONE

Journal Requirements:

"The authors report no conflicts of interest in this work, except KPW which reports research support/donated materials from DuPont Nutrition & Biosciences; Grain Processing Corporation; and Friesland Campina Innovation Centre. KPW’s financial relationships: consulting with or without receiving fees and/or serving on the advisory boards for Circadian Therapeutics, Ltd., Circadian Biotherapies, Inc., and the United States Army Medical Research and Materiel Command – Walter Reed Army Institute of Research."

We note that you received funding from a commercial source: Circadian Therapeutics, Ltd., Circadian Biotherapies, Inc., United States Army Medical Research and Materiel Command – Walter Reed Army Institute of Research.

Within this Competing Interests Statement, please confirm that this does not alter your adherence to all PLOS ONE policies on sharing data and materials by including the following statement: "This does not alter our adherence to PLOS ONE policies on sharing data and materials.” (as detailed online in our guide for authors http://journals.plos.org/plosone/s/competing-interests).  

If there are restrictions on sharing of data and/or materials, please state these. Please note that we cannot proceed with consideration of your article until this information has been declared. 

5. Please ensure that you refer to Figure 3 in your text as, if accepted, production will need this reference to link the reader to the figure.

Reviewers' comments:

Reviewer's Responses to Questions

**Comments to the Author**

1. Is the manuscript technically sound, and do the data support the conclusions?

Reviewer #1: Partly

Reviewer #2: Yes

2. Has the statistical analysis been performed appropriately and rigorously? 

Reviewer #1: Yes

Reviewer #2: Yes

3. Have the authors made all data underlying the findings in their manuscript fully available?

Reviewer #1: Yes

Reviewer #2: Yes

4. Is the manuscript presented in an intelligible fashion and written in standard English?

Reviewer #1: Yes

Reviewer #2: Yes

5. Review Comments to the Author

Reviewer #1: My major concern is the that the "composite" of the sleep wake patterns were created with a mixed of factors that could be controversial . As an example Sleep activity: refers to sleep efficiency (that was not defined), sleep movements (other unit of measure) and nocturnal wake episodes. I saw similar confusion in the other 3 variables which conformed the index. Nevertheless the results were in an appropriate direction.

2. The authors pointed out the interaction of the eating behavior over sleep, but they did not analyze the possibility that sleep patterns could be the predictor factor, or at least mention that they might be interrelated.

Reviewer #2: The original research is novel and value addition to the existing literature. The eating patterns are set very early in life and are getting correlated to sleep pattern is a valuable finding. Parental efforts to regularise it early is also rewarding. The attempt to link it with Gut Microbiota is note worthy. The paper does not capture the breast feeding pattern, weaning foods, its variety as all these factors can also influence eating , sleep pattern and gut microbiota.

6. PLOS authors have the option to publish the peer review history of their article (what does this mean?). If published, this will include your full peer review and any attached files.

Reviewer #1: No

Reviewer #2: **Yes: **Jagmeet Madan

---

## [Decision Letter · Decision Letter 1]

29 Aug 2023

Not simply a matter of parents - Infants’ sleep-wake patterns are associated with their regularity of eating

PONE-D-23-13344R1

Dear Dr. Kurth,

We’re pleased to inform you that your manuscript has been judged scientifically suitable for publication and will be formally accepted for publication once it meets all outstanding technical requirements.

Kind regards,

Takahiro J. Nakamura, Ph.D.

Academic Editor

PLOS ONE

Additional Editor Comments (optional):

Reviewers' comments:

Reviewer's Responses to Questions

**Comments to the Author**

1. If the authors have adequately addressed your comments raised in a previous round of review and you feel that this manuscript is now acceptable for publication, you may indicate that here to bypass the “Comments to the Author” section, enter your conflict of interest statement in the “Confidential to Editor” section, and submit your "Accept" recommendation.

Reviewer #1: All comments have been addressed

Reviewer #2: All comments have been addressed

2. Is the manuscript technically sound, and do the data support the conclusions?

Reviewer #1: Yes

Reviewer #2: Yes

3. Has the statistical analysis been performed appropriately and rigorously? 

Reviewer #1: Yes

Reviewer #2: Yes

4. Have the authors made all data underlying the findings in their manuscript fully available?

Reviewer #1: Yes

Reviewer #2: Yes

5. Is the manuscript presented in an intelligible fashion and written in standard English?

Reviewer #1: Yes

Reviewer #2: Yes

6. Review Comments to the Author

Reviewer #1: (No Response)

Reviewer #2: All the responses by the authors are noted. Appreciate the proactiveness to consider the additional areas for future research.

7. PLOS authors have the option to publish the peer review history of their article (what does this mean?). If published, this will include your full peer review and any attached files.

Reviewer #1: **Yes: **Cecilia Algarin

Reviewer #2: **Yes: **Jagmeet Madan

---

## [Editor Report · Acceptance letter]

27 Sep 2023

PONE-D-23-13344R1 

Not simply a matter of parents - Infants’ sleep-wake patterns are associated with their regularity of eating 

Dear Dr. Kurth:

I'm pleased to inform you that your manuscript has been deemed suitable for publication in PLOS ONE. Congratulations! Your manuscript is now with our production department. 

Kind regards, 

on behalf of

Dr. Takahiro J. Nakamura 

Academic Editor

PLOS ONE